# Analysis of Answers to Queries among Anonymous Users with Gastroenterological Problems on an Internet Forum

**DOI:** 10.3390/ijerph17031042

**Published:** 2020-02-06

**Authors:** Mikołaj Kamiński, Michał Borger, Piotr Prymas, Agnieszka Muth, Adam Stachowski, Igor Łoniewski, Wojciech Marlicz

**Affiliations:** 1Sanprobi Sp.z o.o. Sp. k., 70-535 Szczecin, Poland; 2Faculty of Medicine I, Poznan University of Medical Sciences, 60-780 Poznań, Poland; michalborger.mb@gmail.com (M.B.); piotr.prymas0@gmail.com (P.P.); muth.agnieszka@gmail.com (A.M.); adam.stachowski@onet.pl (A.S.); 3Department of Biochemistry and Human Nutrition, Pomeranian Medical University, 70-204 Szczecin, Poland; sanprobi@sanprobi.pl; 4Department of Gastroenterology, Pomeranian Medical University, 70-204 Szczecin, Poland; marlicz@hotmail.com

**Keywords:** forum, Internet, gastrointestinal ailments, infodemiology, abdominal pain, advice, thread, diarrhea, Poland, bloating

## Abstract

Internet forums are an attractive source of health-related information. We aimed to investigate threads in the gastroenterological section of a popular Polish medical forum for anonymous users. We characterised the following aspects in threads: the main problem of the original poster, declared ailments and rationale of the responses (rational, neutral, harmful or not related to the problem of the original poster). We analysed over 2717 forum threads initiated in the years 2010–2018. Users mostly asked for diagnosis of the problem [1814 (66.8%)], treatment [1056 (38.9%)] and diagnostic interpretation [308 (11.3%)]. The most commonly declared symptoms were abdominal pain [1046 (38.5%)], diarrhea [454 (16.7%)] and bloating [354 (13.0%)]. Alarm symptoms were mentioned in 309 (11.4%) threads. From the total 3550 responses, 1257 (35.4%) were assessed as rational, 693 (19.5%) as neutral, 157 (4.4%) as harmful and 1440 (40.6%) as not related to the user’s problem. The original poster’s declaration of blood in stool, dyspepsia, pain in the abdominal right lower quadrant, weight loss or inflammatory bowel disease was positively related to obtaining at least one potentially harmful response. Advice from anonymous users on Internet forums may be irrational and disregards alarm symptoms, which can delay the diagnosis of life-threatening diseases.

## 1. Introduction

Gastrointestinal ailments are perceived as an embarrassing problem [1]. Almost 45% of men and 33% of women do not inform the physician about heartburn [2], and 50% of the individuals do not disclose problems with faecal incontinence [3,4]. It is reported that women, younger people and patients with the worse condition are more likely not to inform a doctor on their health problem [5]. A lack of trust in confidentiality in a medical office and limited access to public healthcare may convince patients to search for treatment on the Internet [6,7,8].

Up to 90% of the web users look for health-related information [9], and 80% of them perceive the information they found as reliable [10]. The Internet provides the comfort of immediate access to numerous websites associated with the query. The web may be attractive for individuals experiencing minor ailments, who are reluctant to seek help in healthcare professional offices [11]. Moreover, users may consult professional or non-professional peers about their problems [12,13,14,15]. Internet forums provide anonymity which may encourage users to confess the most embarrassing problems. However, the namelessness of peers using the same forum should limit trust in the received answers. Despite this, it is suggested that anonymous health-related advice may significantly affect the decision of the user [16].

Infodemiology (information epidemiology) is an approach to investigating data from e-sources for epidemiological purposes [17,18]. Infodemiology analyses data from social media, search engines and web forums [19]. Previously, analyses on small samples of health-related threads reported good quality of most of the received medical advice [20,21]. However, data on Internet forum threads on gastrointestinal problems are scarce. We hypothesize that analysis of a large number of threads from health forums may provide a unique insight into the most prevalent health problems of the e-community and the accuracy of the advice.

We aimed to analyse threads associated with gastrointestinal complaints on an Internet forum for anonymous users. The investigation involved characterisation of the most prevalent health problems as well as an assessment of answers given by anonymous peers.

## 2. Material and Methods

### 2.1. Data Collection

We analysed threads from the gastroenterological section of a popular Internet Polish health forum for anonymous users. We disclosed the name of the forum to the Editor-in-Chief and reviewers of the manuscript. The name of the forum is not published to not promote the website. In total, forum consists of over 150,000 threads, 2 million posts and is used by over 130,000 registered users. The website has moderators but in our experience the intervention of the administrators is minimal. The forum enables users to initialise a thread both with and without registration and is active and up to date. The registration of an account requires providing an e-mail without any personal data. In Poland, web-scraping is not forbidden by any regulation. The terms of service of the forum also do not forbid web-scraping. Public statements of anonymous individuals such as commentary does not apply to the Intellectual Property Protection Act. The data we collected did not include personal data thus regulations of the General Data Protection Regulation of the European Union do not apply in this case. We scrapped data from all available threads on August 4th, 2018, using *rvest* package of R 3.6.1 (R Foundation, Vienna, Austria) [22]. The collection involves thread URL, user nickname, thread title, initial post and comments. This is a retrospective infodemiology study using data from a forum for anonymous users. Our analysis did not create any risk for the subjects due to anonymity of the data. Therefore, we neither violated the Helsinki Declaration 1964 nor required Ethical Committee approval.

### 2.2. Data Analysis

All threads were analysed after presenting to the coauthors the protocol of the study. The supervisor, medical specialist of gastroenterology (WM), has been consulted about the doubts. We characterised the main declared problems. The problems were categorised as ‘simple’ (clear description of ailment) or ‘complex’ (ambiguous description or many declared problems at once). Moreover, we characterised the type of the initial post as ‘question on treatment’, ‘question on diagnosis’, ‘request for diagnostic interpretation’, ‘request for drug/doctor/healthcare facility recommendation’ (the original poster (OP) declared that the problem is known/diagnosed), ‘request for emotional support or motivation’ and ‘others, not related’. The category ‘others, not related’ contained duplicate posts, advertisements or posts initializing discussion about healthcare or a health condition in general (the OP does not declare any disease or complaint). The initial post could be classified into more than one category. However, we noted all diseases declared by the OPs, as well as complaints and duration of the problem. The working definitions of the 15th most prevalent symptoms are presented in Appendix A. We characterised the advice of the other anonymous users and shortly noted all of them. We performed a descriptive analysis. 

To analyse the factor that decides which initial posts get at least one answer related to the OP problem (which means excluding responses containing advertisement information), we performed logistic regression analysis. The dependent variables were the presence of at least one answer related to the OP problem. The independent variables were the category of the initial post, the complexity of the problem (complex counted as one), the duration of the problem (longer than four weeks counted as one), the presence of each disease and the presence of each symptom. We included all independent variables with a *p*-value at least <0.1 in the multivariate model. We excluded variables with extreme confidence interval bounds (0 to infinite).

We investigated the rationality of users’ advice. We set one of the following categories to each response: rational, the answer is medically justifiable; neutral, the answer will probably not help, but the risk of side effects is minimal; harmful, the respondent omits an alarm symptom and/or the advice is risky for the patient; and not included, the answer does not refer to the OP problem or does not give a proposition for treatment. In case of questions on the interpretation of the diagnostic results or recommendations of healthcare facilities or professionals, answers with the correct interpretation or suitable recommendation were marked as rational while interpretations irrationally downplaying the findings of the diagnosis or suggesting alternative medical specialists instead of a medical professional were characterised as harmful. Other types of answers were marked as neutral. All commentaries to visit medical offices were assessed as rational. Two authors (MK and MB) independently assessed all pieces of advice. Any inconsistencies in the lists were referred to by the senior author (WM).

We performed a logistic regression analysis to assess the characteristics of initial posts that are endangered by harmful advice. The dependent variable was the initial post with at least one piece of harmful advice (coded as one) vs. the initial post with non-harmful piece(s) of advice (coded as null). The independent variables were the category of the initial post, the complexity of the problem (complex counted as one), the duration of the problem (longer than four weeks counted as one), the presence of each disease, and the presence of each symptom. We included all independent variables with a *p*-value at least <0.1 in the multivariate model. We excluded variables with extreme confidence interval bounds (0 to infinite).

We analysed answers with the same content to identify spam. We investigated the duplicate content to seek for potential commercial adverts.

## 3. Results

### 3.1. General Characteristics

Overall, we analysed *n* = 2862 forum threads. We identified *n* = 145 (0.05%) threads, with initial posts classified as off-topic or advertisements which were excluded from further analysis. The number of initial posts according to their type was as follows (percentage of all included threads equal to *n* = 2717): ‘question on treatment’ *n* = 1056 (38.87%), ‘question on diagnosis’ *n* = 1814 (66.76%), ‘request for diagnostic interpretation’ *n* = 308 (11.34%), ‘request for drug/doctor/healthcare facility recommendation’ *n* = 96 (3.53%) and ‘request for emotional support or motivation’ *n* = 129 (4.75%)

Overall, *n* = 1535 (56.5%) OPs provided duration of the problem. In *n* = 854 (31.4%) threads, a complaint lasted at least four weeks, while in *n* = 681 (25.1%) a complaint lasted less than four weeks. As many as *n* = 1306 (48.1%) OPs’ problems were categorised as ‘complex’. 

### 3.2. Declared Symptoms

In total, the users declared *n* = 5192 symptoms. Anonymous users declared the most prevalent symptoms which are abdominal pain [*n* = 1046 (38.5%)], diarrhea [*n* = 454 (16.7%)], bloating [*n* = 354 (13.0%)], nausea [*n* = 307 (11.3%)] and constipation [*n* = 288 (10.6%)] (Table 1). We presented the number of declared locations of abdominal pain in Figure 1. A substantial proportion of the users did not declare abdominal pain location [*n* = 284 (27.2%)]. Moreover, in *n* = 309 (11.4%) threads, users declared at least one alarm symptom [total number (percentage of all analysed threads)]: blood in stool (including tarry stool and stool with fresh blood) [183 (6.7%)], weight loss [94 (3.5%)], dysphagia [18 (0.7%)], ascites [13 (0.5%)], odynophagia [9 (0.3%)] and hemoptysis [7 (0.3%)].

### 3.3. Declared Conditions

In total, the users declared *n* = 716 conditions. The top ten most common declared conditions were gastroenteritis [*n* = 97 (3.6%)], irritable bowel syndrome [*n* = 67 (2.5%)] and hemorrhoids [*n* = 45 (1.7%)] (Table 1).

### 3.4. Posts with Answers

A total of *n* = 1172 (43.1%) posts obtained answers related to the OP problem. The results of univariate logistic regression analysis are presented in Appendix A. In the multivariate regression model, posts with declaration of complaints, blood in stool (OR [95% CI], 1.54 [1.11–2.11]; *p* = 0.01), constipation (1.38 [1.05–1.77]; *p* = 0.02), diarrhea (1.29 [1.04–1.60]; *p* = 0.02), and rectal or anal pain (1.53 [1.00–2.35]; *p* = 0.049), were positively associated with at least one answer, while posts with request for interpretation of the diagnostic results (0.56 [0.42–0.73]; *p* < 0.001) were negatively associated with responses to the initial post (Figure 2). 

### 3.5. Analysis of the Answers

We analysed a total of *n* = 3550 responses. A total of *n* = 316 (8.9%) pieces of advice required assessment by the senior author. After classification of all answers, we found *n* = 1257 (35.4%) rational answers, 693 (19.5%) neutral, 157 (4.4%) harmful and 1440 (40.6%) not related to the problem of the OP. We included translated examples of rational and harmful advice with commentary in Appendix A. Among the rational answers, 615 (48.9%) rational pieces of advice recommended consultation with a physician (either GP or medical specialist). The summary of advice on initial posts with alarm symptoms is presented in Figure 3.

We found that a total of *n* = 87 threads (7.4% of threads with at least one response) had at least one potentially harmful piece of advice for the OP. From the total *n* = 159 threads with declaration of at least one alarm symptom and at least one answer, in *n* = 42 (42/159 = 26.4%) of these threads, the OP obtained at least one harmful piece of advice. The characteristics of declared symptoms and diseases associated with the occurrence of at least one harmful piece of advice in the univariate logistic regression model are presented in Appendix A. In the multivariate model, initial posts with declaration of the occurrence of blood in stool (6.34 [3.59–11.22]; *p* < 0.001), dyspepsia (2.83 [1.06–7.57]; *p* = 0.40), pain in the right lower quadrant (3.35 [1.24–9.10]; *p* = 0.02), weight loss (3.68 [1.60–8.53]; *p* < 0.01) and suffering from hypothyroidism (6.62 [1.13–38.79]; *p* = 0.040) or inflammatory bowel disease (IBD) (4.64 [1.04–20.62]; *p* = 0.044) were independently endangered by harmful advice from anonymous users (Figure 4).

### 3.6. Duplicated Posts

We identified a total of *n* = 16 types of duplicate answers which totally appeared 59 times (1.7%) among answers. Among the unique duplicate answers, *n* = 7 were advertisements of a probiotic brand name which was posted in a total of *n* = 22 answers (Appendix A). Most of the other duplicate posts recommended searching for an answer on a different medical forum or medical website. 

## 4. Discussion

In this infodemiological study, we investigated a large number of forum threads related to digestive system ailments. We carefully analysed the main and side problems of users as well as the rationality of the respondents. This approach enables us to assess the main problems of anonymous forum users with gastrointestinal ailments.

### 4.1. Main Findings

The forum community discussed both recently developed health issues and everyday life problems with chronic illness [23]. Most of the users search for advice concerning diagnosis or treatment of the health problem or diagnostic findings’ interpretation. This does not exclude further consultation with a health professional, rather than seeking for initial advice or to confront the information from a professional [24]. Since a forum is not solely dedicated to emotional problems, only less than 5% of the initial posts users searched for emotional support, which is similar to the previously described Internet forum [14].

Data on the prevalence of gastrointestinal symptoms in Poland are scarce. Ziółkowski et al. reported that the most common abdominal symptoms in a middle-sized Polish city are heartburn/reflux (36%), bloating (31%), dyspepsia (23%) and constipation (13%) [25]. In a large US study, the most prevalent symptoms in the general population were heartburn/reflux (31%), abdominal pain (25%), bloating (21%), diarrhea (20%), constipation (20%) and nausea/vomiting (10%) [26]. The prevalence of symptoms declared by forum users partially reflects the real-world prevalence. It seems that forum users are concerned with symptoms that might be mainly related to common organic diseases such as gastroenteritis as well as functional bowel disorders [27,28]. The abdominal pain was the most commonly reported pain in the upper quadrant and umbilical region which may be related to gastritis, biliary colic, or functional disorders. Interestingly, heartburn was declared only by 5.30% of the OPs. In a previous report, gastroesophageal reflux disease symptoms bothered 27.1–58.0% of the users, and they were the most prevalent abdominal symptoms among Polish adults [25]. Moreover, analysis of Google Ads data revealed that Google users from Poland tend to intensively search for information related to heartburn in comparison with other Western countries [11]. We hypothesise that people may ignore heartburn due to its high prevalence or try to eliminate precipitating food as well as using over-the-counter antacids [29]. In up to 6.74% of the threads, users declared the occurrence of blood in stool. This symptom may be perceived as both embarrassing and ominous. A total of 1.5% of patients older than 34 years report rectal bleeding in general practice in the UK [30]. However, it is reported that blood in stool may occur in 15.5–18.0% of adults during a year, but only a minority of them seek healthcare [31,32]. Therefore, individuals suffering from blood in stool might perceive a forum for anonymous users as a comfortable area for disclosing the problem. Other alarm symptoms were less commonly reported. Currently, there are many efforts to provide online interventions to individuals who present suicidal behaviour on the Internet [33,34]. For instance, the crisis support service for people with suicidal thoughts in the US is provided via both phone hotline and anonymous chat [35]. Similarly, it is worth considering informing the community about alarm symptoms as well as working up similar online interventions for individuals who search for information related to these symptoms. This kind of tool may improve early diagnosis of life-threatening diseases. Using Google Ads may create targeted advertisements of quality websites for those who seek information on alarm symptoms [36]. This may help to direct users to a proper health facility. However, this requires further studies on the efficacy of such campaigns. Only a small portion of the users disclosed a disease they suffered from. This is not surprising because most of the OPs searched for advice on the diagnosis of declared ailments.

We found that declaration of the occurrence of blood in stool, constipation, diarrhea and rectal or anal pain was positively related to obtaining at least one answer from peers. It seems that ailments related to defecation were more likely to get at least one answer. We suppose that disclosing such an embarrassing symptom may encourage other users with similar experiences to respond to the problem. Peers were less likely to interpret diagnostic results which may suggest that anonymous users do not have the proper knowledge to do that.

We assessed 35% of the responses as rational, and almost half of them recommended consultation with a physician, while 4.4% of the responses were harmful. Our outcomes are similar to those of previous reports where a large amount of the advice on health forums is somewhat rational and only a minority is irrational or has a poor quality [20,37]. Importantly, we found that several characteristics of the initial posts might be positively associated with at least one potentially harmful response. It is particularly dangerous for users who may experience alarm symptoms or suffer from a serious condition. Individuals with IBD tend to actively search for information related to their diseases [38,39,40]. Our study suggests that these users may be endangered by obtaining harmful advice. It was previously reported that e-content dedicated to IBD might be a poor source of education [41,42], which may encourage patients to use a complementary medicine method without a scientific background [43,44]. Forum users tend to rely on social media sources and their own experience, but not on scientific sources [45]. For this reason, advice on minor ailments might be somewhat rational, but the poor knowledge may lead to disregarding alarm symptoms. Therefore, the public should be informed about alarm symptoms and encouraged to seek an immediate consultation with a physician. This may help in the early diagnosis of life-threatening diseases. This is particularly important in gastrointestinal malignancies which are characterised by moderate-to-poor long-term prognosis [46]. Such public campaigns could be driven via e-advertisement which was suggested in previous reports [47,48].

We found that approximately 1.7% of all responses were duplicated. In a substantial part of the duplicate posts, anonymous users recommended probiotics or using another website. Previously, it was demonstrated that online forums may be an area of marketing campaigns [49,50]. Pharmaceutical commercials may target both consumers and specialists [51]. It is the responsibility of the e-marketers to not mislead potential consumers on the Internet. Particularly, the e-marketer should not recommend the product on an anonymous forum if users declare an alarm symptom or condition that will not be improved by the advocated product. We do not know the motivation of users who wrote duplicate posts. It could be either commercial or quickly sharing their own experience in similar threads. Nevertheless, this issue requires further dedicated studies.

### 4.2. Strengths and Practical Implications

This is the first comprehensive infodemiology study investigating numerous forum posts related to gastroenterological problems. We analysed all available threads on a selected section of the forum. We characterised the most common symptoms and diseases reported by forum users. We presented some examples of interesting conversations on the forum. A substantial number of pieces of advice were potentially harmful for users. Disregarding alarm symptoms may delay the diagnosis of life-threatening diseases. This is particularly important in gastrointestinal malignancies that are characterized by moderate-to-poor long-term prognosis [46]. Internet forums are poorly investigated, and this study suggests that their content may mislead users. We cannot conclude what decision was made by the users. However, even anonymous advice might be convincing [16]; thus, there is a possibility that some users made the wrong decision. For this reason, the study provides a background for a public discussion on the reliability of this kind of health-related sources. Users should be critical of the information that is not provided by health professionals. The administrators of such forums should implement methods to detect users disclosing alarm symptoms. Prompt direction of users with emergency symptoms to a proper medical facility may prevent delaying the diagnosis. Moreover, physicians should be aware that noncompliance may be related to patients’ discussions on forums [52]. There is a need for further studies investigating real-life consequences of advice from the Internet sources. 

### 4.3. Limitations

We acknowledge several flaws in this study. Firstly, data were retrieved from one anonymous forum for Polish-speaking users. This limits generalisation of the outcomes to the other nations and to other anonymous forums. Secondly, the assessment of the answers was subjective. To restrain the evaluation bias, the pieces of advice were independently analysed by two authors, and inconsistencies were corrected by the senior author. Nevertheless, these evaluations are arbitrary, and this point affects the reproducibility of the study. Moreover, users did not always disclose their general characteristics such as age and gender or characteristics of the complaint and detailed medical history. For this reason, the accuracy of advice and its assessment is limited. Thirdly, the infodemiological study using data from an Internet forum is time-consuming. We were not able to use a natural language processing program viable for this analysis in the Polish language. However, we hope that future analysis of medical forum content may be enhanced by the application of natural language processing methods [35]. Finally, the study does not provide the characteristics of the users. We do not know the motivation of the anonymous users to use the medical forum, their age, place of living, etc.

## 5. Conclusions

The prevalence of ailments declared by the forum users is similar to the real-world prevalence of the symptoms. Advice from anonymous users on Internet forums may be irrational and disregard alarm symptoms, which can delay the diagnosis of life-threatening diseases. The public should be aware of this danger, and this problem requires forum administrators to take action that would promptly direct users who report emergency symptoms to seek medical advice.

## Figures and Tables

**Figure 1 ijerph-17-01042-f001:**
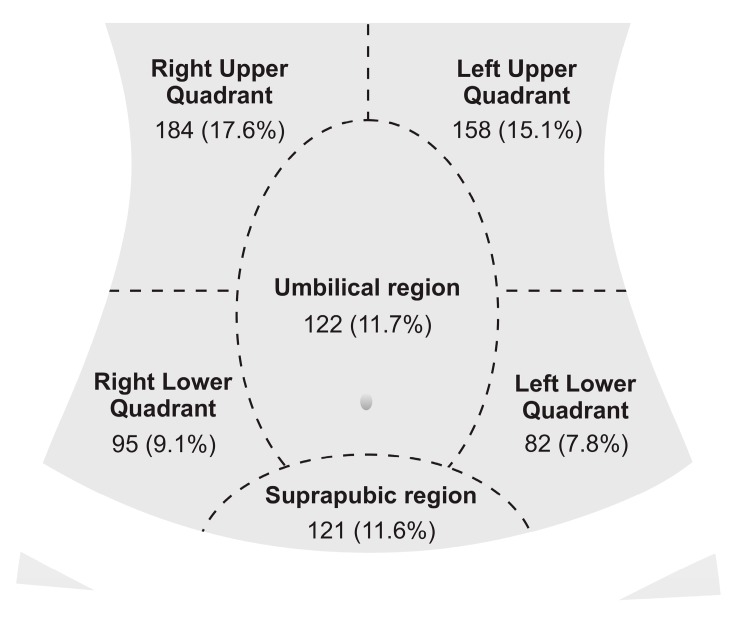
Location of the abdominal pain declared on a gastroenterology forum for anonymous users.

**Figure 2 ijerph-17-01042-f002:**
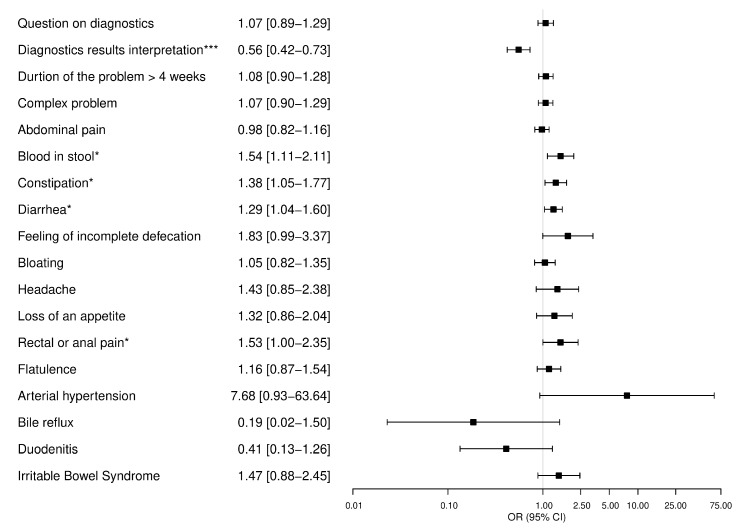
Multivariate logistic regression analysis. Dependent model: obtaining an answer related to the original poster’s problem (coded as one). The results are presented as odds ratio [95% confidence interval]. CI: Confidence Interval, OR: Odds Ratio, * *p*-value < 0.05, *** *p*-value < 0.001.

**Figure 3 ijerph-17-01042-f003:**
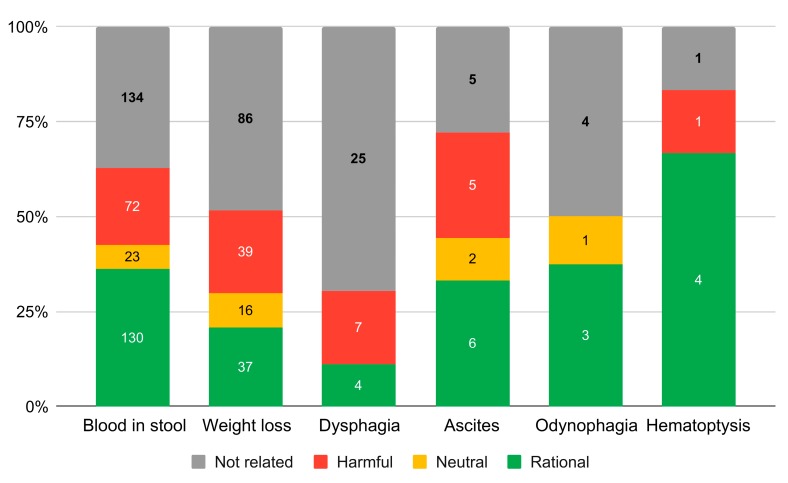
The number of pieces of advice and their characteristics in the initial post about the alarm symptom.

**Figure 4 ijerph-17-01042-f004:**
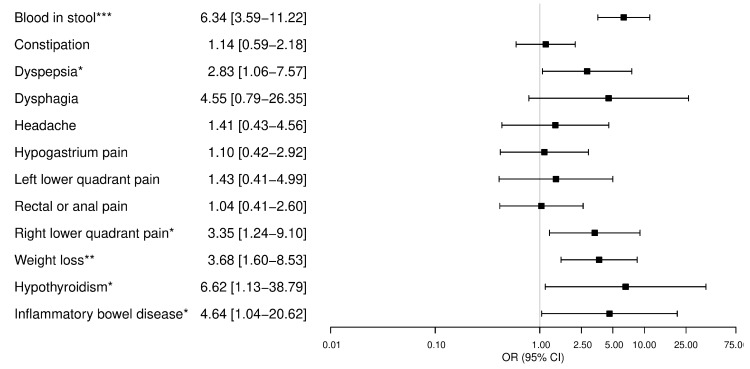
Multivariate logistic regression analysis. Dependent model: the initial post with at least one harmful answer (coded as one). The results are presented as odds ratio [95% confidence interval]. CI: Confidence Interval, OR: Odds Ratio, * *p*-value < 0.05, ** *p*-value < 0.01, *** *p*-value < 0.001.

**Table 1 ijerph-17-01042-t001:** The most prevalent declared ailments and conditions on the gastroenterology section of the Polish Internet forum for anonymous users.

Top	Ailment	Conditions
1.	Abdominal pain	Gastroenteritis
	1046 (38.5%) [20.1%]	96 (3.53%) [13.4%]
2.	Diarrhoea	Irritable bowel syndrome
	454 (16.71%) [8.7%]	67 (2.47%) [9.4%]
3.	Bloating	Haemorrhoids
	354 (13.03%) [6.8%]	45 (1.66%) [6.3%]
4.	Nausea	*Helicobacter pylori* infection
	307 (11.30%) [5.9%]	44 (1.62%) [6.1%]
5.	Constipation	Hiatal hernia
	288 (10.60%) [5.5%]	39 (1.44%) [5.4%]
6.	Flatulence	Inflammatory bowel disease
	256 (9.42%) [4.9%]	30 (1.10%) [4.2%]
7.	Intestinal rumbling	Peptic ulcer disease
	218 (8.02%) [4.2%]	26 (0.96%) [3.6%]
8.	Fatigue	Pregnancy
	189 (6.96%) [3.6%]	22 (0.81%) [3.1%]
9.	Blood in stool	Hypothyroidism
	183 (6.74%) [3.5%]	20 (0.74%) [2.8%]
10.	Vomit	Duodenitis
	153 (5.63%) [2.9%]	19 (0.70%) [2.7%]
11.	Heartburn	Food allergy or intolerance
	144 (5.30%) [2.8%]	16 (0.59%) [2.2%]
12.	Belching	Anxiety disorders
	136 (5.01%) [2.6%]	15 (0.55%) [2.1%]
13.	Rectal or anal pain	Overweight
	97 (3.57%) [1.9%]	14 (0.52%) [2.0%]
14.	Weight loss	Biliary system diseases
	94 (3.46%) [1.8%]	12 (0.44%) [1.7%]
15.	Loss of appetite	Parasites infection
		Coeliac disease
	89 (3.28%) [1.7%]	11 (0.40%) [1.5%]

Data is presented as number and percentage of the total number of analysed threads in round brackets, and a percentage of the total number of declared symptoms or conditions in square brackets. Data is presented as the total number of users who declared location of the abdominal pain [*n* = 1046 (100%)] and percentage of the total number of users who declared abdominal pain.

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
