# Peer review of "Analysis of Answers to Queries among Anonymous Users with Gastroenterological Problems on an Internet Forum"

_ijerph, 2020, doi:10.3390/ijerph17031042_

Round 1

Reviewer 1 Report

Sir, 

first at all, thank you for the opportity to review this very interesting paper on the potential detrimental impact of social media consultation on the management of clinical conditions.

The paper is quite original, both in its design and in its composition: the novelty of the main theme, the overall quality of the study collectively suggest a quick publication on IJPERPH.

Still, I have two concerns that should be preventively managed by the Authors:

1) please report the reference of the internet forum you assessed, explaining whether you had preventively obtained the authorization by the managers of the internet forum to fetch their data. In case you did no require such authorization, please explain in futher details: I'm absolutely aware that you explained how "In Poland, web-scraping is not forbidden by any regulation", but we are dealing with the critical issue of intelletual property of the data you analyzed. Similarly, as Poland is a member state of the EU, please explain whether the analyses you performed are compliant with the GDPR framework (https://eur-lex.europa.eu/legal-content/PL/TXT/HTML/?uri=CELEX:32016R0679&from=IT).

2) as you clearly explained in the methods section, you accurately reviewed the complaints of the internet forum members in order to fetch them in a series of categories (Table 1). In order to improve the reproducibility of your assessment, please include in the "supplementary material" a table with the working definition of the signs/symptoms.

I'm certain that the Authors will fix in a short term both concerns, allowing the senior Editors to evaluate the quick publication that, I'm certain, this study eventually deserves.

Reviewer 2 Report

Thank you for the opportunity to review the paper “ Analysis of threads on a medical forum for anonymous users with gastroenterological problems: a retrospective infodemiology study”. In this paper, authors analyze discussions from a gastroenterological section of an anonymous medical forum. The authors assess the character of suggestions provided by the users and qualify the fraction of reasonable and potentially harmful advices.

Overall, I like the idea behind the paper. Even before seeing the results, I was already curious, what will be the outcome. Hence, I believe that this paper will be of interest to the general public. However, there are some issues, which remain to be resolved. Therefore, I suggest minor revision before a paper can be published.

Below are my comments:

I am concerned about how representative is the data source. What is stated in the paper is that the data was scrapped from “popular polish health forum for anonymous users”. What is the count (or estimation) of users at this forum? Are there other similar internet-communities in Poland? How can they be compared with the chosen forum (more/less popular, more/less specific)? If I am a general Polish citizen looking for health advice on the web, what are my chances to come to the chosen forum? What is the moderation/archiving policy on the chosen forum? Is it possible that answers seen by the topic starter (hours and days after OP) are not the same as answers collected by authors (years after OP).

Figs. 2 and 4. Are not explained well enough. What are these numbers actually mean? What is the “odds ratio”? I cannot infer any information from these figures. With the current exposition, these figures are not useful, unfortunately.

The presentation of fitting results in a form (X [Y-Z]; P=W), which is used e.g. at the lines 164 – 167, is opaque as well. What do these values mean?

Reviewer 3 Report

This article by Mikolaj Kaminski et al. described an extensive analysis of threads analysed from a medical forum. This well-written article is very interesting to highlight limit of the clinical management of gastrointestinal symptomatologies. Some minor issues remains: 

General comment : 

Prefer use passive form in place of active form

Specific comment : 

Abstract :

Introduction is lacking in the abstract.

"gastrointestinal ailments declared on the forum patially reflect real-worl prevalence". Please reword or give precisions.

Introduction : 

Is non-information of physician related to age/sex/socieconomic characteristics?

Line 35-36 : this sentence need reference

after line 46 : the presentation introduced blank page

Methods : 

Line 62 : give referrence to the analysed website and decribed its structure more extensively (moderation of the threads from the administrators for example ?)

Line 70 : give precision about the Ethical committee approval.

Results : 

Table 1 : present the Table 1 as respective proportion and number of the declared ailment and declared conditions.

Figures 1 and 2 : Give explanation of *, ** and ***. Moreover a representation of the interval for univariate analysis could be interesting to add to the figure (as a diamond for example ?)

Line 192 : are consequence or response to harmfull comments analysed ? I could be very interesting.
